# Microbiome Structures and Beneficial Bacteria in Soybean Roots Under Field Conditions of Prolonged High Temperatures and Drought Stress

**DOI:** 10.3390/microorganisms12122630

**Published:** 2024-12-19

**Authors:** Sandeep Gouli, Aqsa Majeed, Jinbao Liu, David Moseley, M. Shahid Mukhtar, Jong Hyun Ham

**Affiliations:** 1Department of Plant Pathology and Crop Physiology, Louisiana State University Agricultural Center, Baton Rouge, LA 70803, USA; sgouli@agcenter.lsu.edu; 2Department of Biology, University of Alabama at Birmingham, 3100 Science & Engineering Complex–East Science Hall, 902 14 Street South, Birmingham, AL 35294, USA; amajeed@clemson.edu (A.M.); jinb2112@uab.edu (J.L.); 3Department of Genetics & Biochemistry, Clemson University, 105 Collings St. Biosystems Research Complex, Clemson, SC 29634, USA; 4Dean Lee Research & Extension Center, Louisiana State University Agricultural Center, Alexandria, LA 71302, USA; dmoseley@agcenter.lsu.edu

**Keywords:** soybean rhizosphere, root endosphere, microbial community

## Abstract

Drought stress has a significant impact on agricultural productivity, affecting key crops such as soybeans, the second most widely cultivated crop in the United States. Endophytic and rhizospheric microbial diversity analyses were conducted with soybean plants cultivated during the 2023 growing season amid extreme weather conditions of prolonged high temperatures and drought in Louisiana. Specifically, surviving and non-surviving soybean plants were collected from two plots of a Louisiana soybean field severely damaged by extreme heat and drought conditions in 2023. Although no significant difference was observed between surviving and non-surviving plants in microbial diversity of the rhizosphere, obvious differences were found in the structure of the endophytic microbial community in root tissues between the two plant conditions. In particular, the bacterial genera belonging to Proteobacteria, *Pseudomonas* and *Pantoea*, were predominant in the surviving root tissues, while the bacterial genus *Streptomyces* was conspicuously dominant in the non-surviving (dead) root tissues. Co-occurrence patterns and network centrality analyses enabled us to discern the intricate characteristics of operational taxonomic units (OTUs) within endophytic and rhizospheric networks. Additionally, we isolated and identified bacterial strains that enhanced soybean tolerance to drought stresses, which were sourced from soybean plants under a drought field condition. The 16S rDNA sequence analysis revealed that the beneficial bacterial strains belong to the genera *Acinetobacter*, *Pseudomonas*, *Enterobacter*, and *Stenotrophomonas*. Specific bacterial strains, particularly those identified as *Acinetobacter pittii* and *Pseudomonas* sp., significantly enhanced plant growth metrics and reduced drought stress indices in soybean plants through seed treatment. Overall, this study advances our understanding of the soybean-associated microbiome structure under drought stress, paving the way for future research to develop innovative strategies and biological tools for enhancing soybean resilience to drought.

## 1. Introduction

Drought stress is one of the most important environmental factors that significantly impact agricultural productivity, leading to a potential yield loss of 50% [1,2]. This can have severe implications for global feed security. The second most widely cultivated crop in the United States is soybeans, representing 32% of the cultivated land. However, under drought conditions, soybeans can experience a reduction in yield of up to 100% [1]. As the crisis of global climate change continues to escalate, the frequency and severity of drought events are intensifying, presenting a greater threat to crop yields and overall agricultural sustainability. Consequently, understanding the mechanisms underlying plant responses to drought stress is imperative for developing resilient soybean varieties and implementing effective management strategies. Utilizing beneficial microbes has emerged as a promising approach to enhance crop resilience against environmental stress, including drought [3].

Assessing the microbial diversity and population of both endophytes and the rhizosphere in crop plants across various environments is crucial to harnessing their potential use in enhancing plant growth and resilience against both biotic and abiotic stress [3]. Microbiome studies have highlighted the pivotal role of plant-associated microbes, particularly in enhancing legume resilience to drought and heat stress. Under these conditions, both the rhizosphere and endophytic microbiomes undergo significant changes, with beneficial microbial populations adapting and intensifying their interactions with plants to support survival [4,5]. These microbes produce phytohormones like abscisic acid (ABA), osmolytes, and antioxidants, mitigating the adverse effects of water scarcity and high temperatures. Additionally, beneficial microbes such as *Pseudomonas*, *Bacillus*, *Rhizobium*, Acinetobacter, and several others improve root architecture, increase water uptake and facilitate nitrogen fixation, ensuring nutrient availability even under stress conditions [6,7]. Microbial biofilms further enhance drought tolerance by promoting soil aggregation and moisture retention, while heat-shock proteins and stress-responsive gene activation mediated by microbes protect plants from cellular damage [8,9,10,11]. Using these microbiome-plant interactions offers a powerful strategy for developing drought- and heat-tolerant legume cultivars, which is critical for sustainable agriculture amidst climate change.

The progress in high-throughput sequencing and bioinformatics tools allows for the evaluation of operational taxonomic units (OTU), amplicon sequence variants (ASV), or species, along with their respective abundances [12]. Leveraging correlation-based and graphical models, among other methods, co-occurrence network analysis is applied to illustrate microbial relationships within diverse spatiotemporal niches [13]. Meanwhile, network topological features, i.e., modularity and connectivity, and several of such network parameters can serve as indicators of significant nodes in the network [14]. These encompass degree, indicating the number of connections a node possesses; betweenness, representing the fraction of the shortest paths passing through a node, among other centrality measures [15]. Recently, weight k-shell decomposition network analysis was shown to be more effective in discovering fast information-spreading nodes [16].

The primary objective of this study was to investigate the microbiome structure of soybean roots under natural drought and heat stresses in the field, particularly during this year’s severe environmental growing season. Additionally, we isolated and identified soybean-associated bacteria from drought-stressed soybean plants and tested their beneficial effects on soybean growth under drought stress with the hypothesis that those bacteria would enhance the drought tolerance of soybean plants.

## 2. Materials and Methods

### 2.1. Plant Sample Collection and Preservation

During the 2023 growing season, most regions of Louisiana experienced extreme weather conditions of high temperatures and drought for an unprecedentedly extended period. From these abnormally hot and dry weather conditions, some of the soybean fields at the Dean Lee Research Station of the LSU AgCenter in Alexandria, Louisiana, exhibited devastating damages, and most of the plants could not survive, causing a more than 90% yield loss (Figure 1). As there were no healthy plants at a similar growth stage cultivated in nearby well-irrigated plots, soybean plants were collected from two damaged plots only. However, both surviving and non-surviving plants were sampled from each plot for comparative analysis. Moreover, the application of plant growth-promoting bacteria (PGPB) as biostimulants has emerged as a promising strategy to mitigate drought stress effects in agricultural systems. As a result, we conducted greenhouse experiments using bacteria isolated from healthy soybean plants in a drought-stressed field adjacent to the microbiome study site but at a different time and plant growth stage.

Soybean plant samples were collected on 29th August 2023 from two neighboring plots (Plot A and Plot B) at the Dean Lee Research Station, Alexandria, Louisiana (31.11° N, 92.24° W), which were severely damaged from the extended period of high temperatures and dry weather conditions during the 2023 growing season (Figure 1 and Figure 2). Five surviving and five dead plants were collected at the R6-R7 stage (~4 months after planting) from both Plot A and Plot B, totaling ten plant samples per plot. Plot A experienced severe drought stress, resulting in the survival of only a few green plants, while Plot B exhibited less severe drought stress with more green plants than Plot A (Figure 1). Soil types in both plots included latanier silty clay loam at the front and moreland clay at the back. Tillage was conducted in March, and planting was conducted on 4 May 2023 for both plots. Different crop varieties were cultivated in each plot: AG49XF3 (Bayer CropScience) for Plot A and P5554RX (Progeny Ag) for Plot B. The collected plant and soil samples were carefully encased in large plastic bags and transported to the laboratory, where they were temporarily stored in a cold room at 4 °C before processing.

### 2.2. Field Management

#### 2.2.1. Insecticide and Herbicide Applications

In both plots, insecticides were applied on two dates: Moccasin MTZ (United Phosphorous, King of Prussia, PA, USA) on 17 July 2023, at a rate of 0.41 mL/m^2^, and Endigo (Syngenta, DE, USA) on 24 July 2023, at a rate of 0.033 mL/m^2^. Herbicide applications were administered on 8 May 2023 with Varsity (Innvictis Crop Care, Boise, ID, USA) at a rate of 0.015 mL/m^2^, Derive (Innvictis Crop Care, Boise, ID, USA) at a rate of 0.045 mL/m^2^, Zidua (BASF Agricultural Solutions, Geismar, LA, USA) at a rate of 0.018 mL/m^2^, and Fever (Innvictis Crop Care, Boise, ID, USA) at a rate of 0.234 mL/m^2^. Charger Max (WinField United, Shoreview, MN, USA) was applied at a rate of 0.146 mL/m^2^ on 9 May 2023 and 15 June 2023, Sentris (BASF Agricultural Solutions, LA, USA) at a rate of 0.06 mL/m^2^. Engenia (BASF Agricultural Solutions, Geismar, LA, USA) at a rate of 0.09 mL/m^2^ and Roundup Powermax (Bayer Crop Science, Chesterfield, MO, USA) at a rate of 0.23 mL/m^2^ were administered.

#### 2.2.2. Fungicide and Fertilizer Applications

On 30 June 2023, a fungicide, Stratego Yield (Bayer CropScience, Chesterfield, MO, USA), was applied at a rate of 0.034 mL/m^2^ g in both fields, supplemented with a 0.25% nonionic surfactant. The fertilizer used in both fields was a composition of N-P-K at 0-18-36.

### 2.3. DNA Sample Preparation

For DNA extraction from the rhizospheric soil, the soil closely attached to the plant roots was collected in 50 mL falcon tubes using a sterile spatula and gloves to prevent contamination. Each sample was labeled appropriately according to the status of each plant sample, and 0.25 g of the collected soil sample were used for DNA extraction. The DNA extraction was performed using the DNeasy PowerSoil Pro Kit (Qiagen GmbH, Hilden, Germany) following the manufacturer’s instructions. After DNA extraction, the quantity and quality of each DNA sample were assessed using a NanoDrop 1000 Spectrophotometer (Thermo Scientific, Wilmington, DE, USA).

For DNA extraction from the inner part of plant roots (endosphere), the plant root samples that remained after collection of the rhizospheric soil were surface-sterilized by immersing them in 70% ethanol for 30 s twice and subsequently rinsing with double-distilled water for 30 s four times to ensure the removal of microbes on the surface of root tissue. Under aseptic conditions and using sterile scissors, the surface-sterilized roots were cut into pieces (~1 to 2 mm thick) to access the inner endospheric part. DNA extraction of the plant tissue samples for the endosphere was conducted utilizing a DNeasy Plant Mini Kit (Qiagen GmbH, Hilden, Germany) following the manufacturer’s instructions.

Subsequently, the DNA samples extracted from the rhizosphere and endosphere samples were quantified using a NanoDrop 1000 Spectrophotometer (Thermo Scientific, Wilmington, DE, USA). Extracted DNA was then sent for 16S rDNA sequencing.

### 2.4. Library Construction and High-Throughput DNA Sequencing of 16S rDNA

The V4 regions of bacterial 16S rDNA were amplified using the primers 515F (5′-GTGYCAGCMGCCGCGGTAA-3′) and 806R (5′-GGACTACNVGGGTWTCTAAT-3′) designed in published protocols [17,18]. Thirty nanograms of isolated DNA were used as the template for the PCR amplification reaction and subsequent library construction. The resultant products were purified using Agencourt AMPure XP beads (Beckman Coulter, Indianapolis, IN, USA) and subsequently measured using an Aligent 2100 bioanalyzer (Agilent, Santa Clara, CA, USA) to determine their size and concentration. Samples that passed the quality control were sequenced on the DNBSEQ-G99 platform (MGI Tech, Shenzhen, China) using a 300bp paired-end strategy.

### 2.5. Raw Data Import, Quality Checking, and ASV Feature Table Construction

Raw paired-end reads (FASTQ) from the original DNA fragments were imported in Qiime2 v 2023.2 software [19]. Paired-end reads for all 20 samples for the dataset Endosphere (root) and 20 samples for rhizosphere (soil) were imported using the manifest file method. Quality filtering, denoising, and chimeric sequence removal were carried out using the DADA2 denoising method. To remove low-quality regions of the sequences, the --p-trunc-len parameter was used to truncate each sequence at position 253 in forward and reverse reads. This DADA2 pipeline generated a FeatureTable (frequency), which contains counts (frequencies) of each unique sequence in each sample in the dataset, and a FeatureData (sequence), which maps feature identifiers in the feature table to these sequences.

### 2.6. Taxonomy Assignment

To explore the taxonomic composition of the samples, a pre-trained Naïve Bayes classifier and q2-feature classifier plugin were used to assign likely taxonomies to the sequences. This classifier, downloaded from the qiime2 data resources page, was trained on the SILVA OTUs from the V4 (515F/806R) region of sequences [20]. Taxa bar plots were generated using an R package microViz (V. 0.12.0) to visualize the taxonomic composition of each sample and group at the phylum, family, and genus classification levels [21]. Bar plots were used to visualize OTUs’ relative abundance. All nonbacterial OTUs’ sequences were filtered out using the feature-table-filtering method in Qiime2.

### 2.7. Diversity Analysis

To assess the alpha diversity, three different metrics were calculated: “Evenness” estimated the species abundance; “Observed OTU” estimated the number of unique OTUs found in each sample; the “Shannon index” accounted for both richness and evenness. The Shannon index value ranges from 0 to 1. Lower values indicate high diversity and higher index values lower diversity. These alpha diversity metrics were calculated using the phyloseq function ‘estimate_richness’, and to visualize diversity results, boxplots are generated using the “qiime2R” and “ggplot2” libraries in R [22,23,24]. For beta diversity analysis, principal coordinate analysis (PCoA) was performed. PCoA is an unconstrained method, but it does require a distance matrix. In an ecological context, a distance (or, more generally, a “dissimilarity”) measure indicates how different a pair of (microbial) ecosystems are. This can be calculated in many ways. For this study, weighted UniFrac distance, unweighted UniFrac distance, and generalized UniFrac, “gunifrac, were selected to generate a PCoA curve to measure the dissimilarity coefficient between pairwise samples, which are phylogenetic measures used extensively in recent microbial community sequencing projects. An R package microViz (V.0.12.0) was used to generate these plots [21]. The UniFrac family of methods was employed to determine dissimilarities. The approach considers the phylogenetic relatedness of taxa/sequences in samples. Conversely, unweighted UniFrac dist_calc(dist = “unifrac”) disregards the relative abundance of taxa and solely highlights their presence (detection) or absence. This renders it particularly sensitive to rare taxa, sequencing artifacts, and abundance filtering choices. For assessing dissimilarities, on the other hand, weighted Unifrac, denoted as “wunifrac”, places (potentially excessive) emphasis on highly abundant taxa. The generalized UniFrac, labeled as “gunifrac”, achieves a balance between the extremes of unweighted and weighted UniFrac.

### 2.8. Co-Occurrence Microbial Network Analysis

An R package called ggClusterNet was used to make networks for endosphere (root) and rhizosphere (soil) datasets [25]. An integrated function network.2 of ggClusterNet was used for microbial network data mining and visualization. Briefly, to calculate network correlation, the Spearman method was used; 0.9 correlation and 0.05 *p*-value thresholds were used to filter the microorganism table. The model_maptree2 layout was used for the microbial network. Two types of networks were calculated: (1) a global network, including all OTU from all samples; (2) an individual network for each of four groups from endosphere (root) and rhizosphere (soil) datasets.

### 2.9. Isolation and Characterization of Beneficial Bacteria from Soybean Plants Under Drought Stress Conditions

Soybean plants in the vegetative stage (V5-V6) were sampled from a field adjacent to a microbiome study site at the Dean Lee Research Station in Alexandria, Louisiana, under drought stress conditions (Appendix A). Healthy plants were selected from a population having both healthy and stressed plants for the isolation of soybean-associated bacteria in roots. To capture a comprehensive range of bacterial species, both rhizospheric soil and endospheric (root) samples were collected. For rhizospheric bacterial isolation, excess soil was gently removed from the roots, leaving only the soil closely attached to the root surface. The roots with attached soil were placed in a 15 mL tube containing 10 mL of buffer solution (10 mM MgCl_2_). The sample was vortexed to detach the soil from the roots, isolating the rhizospheric soil. The rhizospheric soil was stored at −20 °C until further processing. A dilution series was prepared with the buffer, and samples were spread onto Nutrient Agar plates (1 g/L beef extract, 2 g/L yeast extract, 5 g/L peptone, 5 g/L sodium chloride, 15 g/L agar, and 1 L distilled water) containing cycloheximide (40 µg/mL) to isolate bacteria only. For endospheric bacterial isolation, root samples were thoroughly washed with distilled water, followed by 70% ethanol, and rinsed again with sterile double-distilled water (ddH_2_O). The roots were then cut and crushed in a mortar with 10 mL of buffer (10 mM MgCl_2_ ). The mixture was diluted and spread onto Nutrient Agar plates amended with cycloheximide (40 µg/mL). Plates containing rhizospheric or endospheric bacterial samples were incubated overnight at 30 °C. The following day, bacteria exhibiting mucoidal colony morphology on the agar plates were selected. Single colonies were transferred to Luria–Bertani Agar (LBA; 10 g/L tryptone, 10 g/L sodium chloride, 5 g/L yeast extract, and 17 g/L agar) plates and incubated overnight. Mucoidal bacteria were confirmed by visual inspection and selected for further characterization. Bacteria were screened for their plant growth-promoting traits. Siderophore production was assessed on Chrome Azurol S (CAS) agar plates prepared using blue dye (Solution 1: 1.2 g CAS per liter ddH_2_O, Solution 2: 0.27 g FeCl_3_·6H_2_O per liter of 10 mM HCl, Solution 3: 1.825 g HDTMA per liter ddH_2_O), mixture solution (MM9 salt solution: 30 g KH_2_PO_4_, 50 g NaCl, 100 g NH_4_Cl per liter ddH_2_O with 200 g glucose, 166.67 g NaOH, and 111.11 g casamino acid per liter ddH_2_O), and final solution (37.929 g piperazine-N,N′-bis(2-ethanesulfonic acid) PIPES per liter of mixture containing 11.76% MM9 salt solution and 88.24% ddH_2_O, with 17.7 g bacto agar per liter ddH_2_O) with positive results indicated by an orange halo around the colonies [26,27]. Nitrogen-fixing bacteria were identified by their fast and dense growth on Jensen’s nitrogen-free agar media (20 g/L sucrose, 1 g/L dipotassium phosphate, 0.5 g/L magnesium sulphate, 0.5 g/L sodium chloride, 0.1 g/L ferrous sulphate, 0.005 g/L sodium molybdate, 2 g/L calcium carbonate, and 15 g/L agar) [28,29]. Phosphate-solubilizing bacteria were screened on Pikovskaya’s agar (0.5 g/L yeast extract, 10 g/L dextrose, 5 g/L calcium phosphate, 0.5 g/L ammonium sulphate, 0.2 g/L potassium chloride, 0.1 g/L magnesium sulphate, 0.0001 g/L manganese sulphate, 0.0001 g/L ferrous sulphate, and 15 g/L agar), with positive isolates forming a clear halo after 7–8 days of incubation at 30 °C [30]. Bacteria exhibiting beneficial traits were preserved in 30% glycerol stock at −80 °C for future use.

### 2.10. Identification of Soybean-Associated Candidate Beneficial Bacteria Through 16S Ribosomal RNA Gene Amplification and Sequencing

Bacterial isolates preserved in 30% glycerol at −80 °C were grown overnight on LB agar plates. The bacterial isolates were then cultured in LB broth overnight. Genomic DNA was extracted from these liquid culture isolates using the Zymo research Quick-DNA miniprep kit (Genesee Scientific, El Cajon, CA, USA) following the manufacturer’s instructions. The 16S rDNA was PCR-amplified using DNAEngine Peltier thermal cycler (Bio-Rad, Hercules, CA, USA) with bacteria-specific forward primer fD1 (5′-CCGAATTCGTCGACAACAGAGTTTGATCCTGGCTCAG-3′) and reverse primer rD1 (3′-CCCGGGATCCAAGCTTAAGGAGGTGATCCAGCC-5′) [31]. Each PCR reaction contained a total of 50 μL reaction mixture comprising 5 μL of 10X PCR (10X paq) buffer, 1 μL of 10 mM dNTPs, 1 μL of DMSO, 2 μL each of forward and reverse primers, 0.4 μL of Paq5000 DNA polymerase (Agilent Technologies, Cedar Creek, TX, USA), 36.6 μL of sterile double-distilled water, and 2 μL of DNA template. The PCR program included an initial denaturation at 95 °C for 5 min, followed by 35 cycles of denaturation at 95 °C for 2 min, annealing at 42 °C for 30 s, and extension at 72 °C for 4 min. A final extension was performed at 72 °C for 20 min, with a holding temperature of 4 °C for temporary storage of the PCR reaction. A 10 μL aliquot of the PCR product was subjected to 1% agarose gel electrophoresis in 1X Tris-Borate-EDTA (TBE) buffer with ethidium bromide at 130 V for approximately 45 min using a Thermo Fisher electrophoresis system (Thermo Fisher Scientific, Waltham, MA, USA). The gel was viewed under Spectroline™ Select™ Series UV Transilluminators (Thermo Fisher Scientific, Waltham, MA, USA). The PCR products were then purified using the SIGMA-ALDRICH GenEluteTM PCR clean-up kit (MilliporeSigma, Saint Louis, MO, USA) following the manufacturer’s instructions. The purified PCR products were sent to the Psomagen sequencing facility for 16S rRNA gene sequencing. After receiving the sequencing data, the sequences were checked for homology to bacteria-specific genes using the BLAST program on the NCBI-BLAST website (http://blast.ncbi.nlm.nih.gov/Blast.cgi, accessed on 7 July 2024). The bacteria were then identified based on the BLAST results.

### 2.11. Evaluating Bacterial Isolates as Biostimulants to Enhance Soybean Growth Under Drought Stress

#### 2.11.1. Experiment Setup, Bacterial Preparation, and Design for Drought Stress

Bacterial isolates preserved at −80 °C were streaked onto LBA plates and incubated overnight at 30 °C. A single loopful of bacteria from the LBA plates was transferred into flasks containing autoclaved Luria–Bertani (LB) broth and incubated overnight in a shaking incubator at 30 °C. The optical density (OD) of each bacterial culture was measured using a spectrophotometer to ensure uniform bacterial concentrations. The dilution formula, C1V1 = C2V2, was used to adjust the volume of each bacterial culture to achieve equal concentrations across all isolates. A carboxymethyl cellulose (CMC) solution (0.2%) was then added to reach the required volume of the bacterial suspension. Surface-sterilized soybean seeds were treated by mixing 2 parts bacterial suspension (mL) with 3 parts seeds (g). The treated seeds were air-dried overnight in preparation for planting.

For each treatment, thirty soybean seeds were sown in medium-sized pots (5 × 5 × 6 inches) filled with a loamy soil and sand mixture (1:1 ratio, pH 6.8) following a completely randomized design. After germination, 15 seedlings were retained in 5 pots, with each pot containing 3 seedlings. Soil moisture was monitored periodically using a moisture meter to maintain a humidity of 35–40% under normal watering conditions and 10–15% during drought-stressed conditions. Soybean plants were watered regularly for one week to ensure healthy seedling emergence before initiating drought stress. After this period, plants were left unwatered for 3–4 days until the soil moisture content reached 10–15%, marking the onset of drought stress. Seven days after the drought stress initiation, data were collected and analyzed using one-way ANOVA and Tukey’s test in RStudio (Version 2024.04.2+764). In total, 12 treatments were evaluated for their effects on soybean growth under drought conditions: five bacterial isolates from the rhizosphere (DRS1–DRS5) and the endosphere (DES1–DES5), respectively, CMC-treated control, and no treatment control.

#### 2.11.2. Determination of Growth-Related Parameters

Root and shoot lengths were measured using a calibrated ruler for precise measurements. Fresh weights of roots and shoots were determined using an analytical balance. Roots were gently washed to remove soil, and both roots and shoots were dried at 55 °C for 48 h to determine dry biomass. The water content of roots and shoots was calculated using the following established formula [32].

Shoot Water Content (%) = (SFW − SDW)/SFW × 100
(SFW = Shoot Fresh Weight, SDW = Shoot Dry Weight),

Root Water Content (%) = (RFW − RDW)/RFW × 100
(RFW = Root Fresh Weight, RDW = Root Dry Weight).

#### 2.11.3. Determination of Drought Stress Index (DSI) and Chlorophyll Content

DSI was determined using a 6-point visual assessment scale adapted and modified from published protocols (Appendix A) [33,34]. Ten trifoliate leaves were randomly selected from each treatment and scored based on leaf phenotype: 1: healthy, 3: slight loss of vigor with no leaf rolling, 5: partial leaf rolling, 7: severe leaf rolling, 9: dried leaf, and 10: fully dried leaf. Chlorophyll content was measured using a Chlorophyll Meter SPAD-502 (Spectrum Technologies, Aurora, IL, USA). Ten plants per treatment were analyzed, and readings were taken from fully expanded trifoliate leaves. The average of 10 Soil Plant Analysis Development (SPAD) readings was used to determine chlorophyll content.

## 3. Results

### 3.1. Alpha-Beta Diversities and Taxa Bar Plots

We analyzed the alpha and beta diversities of both rhizospheric and endospheric bacterial communities. As we did not find any significant differences in beta diversity among the rhizosphere samples regardless of their surviving status and plot condition, our analysis of the bacterial community shifted to the root endosphere. Here, we observed significant differences in beta diversity between drought-surviving and non-surviving soybean root tissues in both Plot A and Plot B (*p*-value = 0.001, permanova, pseudo-F test statistic). Pairwise permanova results further confirmed these distinctions. Notably, alpha diversity analysis revealed higher species diversity and evenness within the groups. A closer scrutinizing of taxa bar plots identified seven genera (*Pseudomonas*, *Pantoea*, *Streptomyces*, *Micrococcaceae*, *Luteimonas, Variovorax*, and *Bacillus*) as being permanently abundant in either surviving or dry/dead plants from both fields (Figure 3 and Figure 4). These findings led us to hypothesize that these bacterial genera might play a crucial role in enhancing soybean tolerance to severe drought stress. A comprehensive bar graph visually illustrates the abundance of these bacterial genera in surviving plants except for *Streptomyces*, which is abundant in dry/dead plants in both fields (Figure 4) [35]. Among the seven major bacterial genera, Gram-negative bacteria *Pseudomonas* was predominant in surviving roots from both plots, while *Pantoea* was conspicuously predominant only in those from Plot A, which had more severe stress damage (Figure 4). In contrast, *Streptomyces* was the genus that was most dominant over other bacterial genera in death root tissues from both plots (Figure 4).

### 3.2. Co-Occurrence Networks Analyses

Network analyses are crucial in deciphering co-occurrence patterns across microbial taxa within complex communities. They also facilitate determining positive and negative interactions among diverse taxa [25]. Here, we constructed two sets of co-occurrence networks: endophytic and rhizospheric networks (Figure 5). Within each category, four networks were developed, namely, surviving plot A, non-surviving plot A, surviving plot B, and non-surviving plot B. In total, eight co-occurrence networks were constructed using the significant correlations among phyla (Spearman’s correlation coefficient r > 0.9, *p* < 0.05). Each network exhibited varied network features (Figure 5, Table 1). In general, rhizospheric networks consist of a greater number of nodes, edges, and clusters compared to all endophytic networks, with the highest numbers observed in surviving plot A. Conversely, among endophytic networks, non-surviving plot A displayed the highest number of nodes, edges, and clusters (Figure 5). These findings suggest the existence of varied levels of OTU complexity in endophytic and rhizospheric networks in plot A. These network features are consistent with the average degree of each network. Intriguingly, non-surviving plot A of endophytic networks showed the highest numbers of negative edges, indicating the existence of both synergistic and antagonistic relations among diverse OTUs. The edge density, a network characteristic that represents the proportion of possible relationships in the network, signifies the intricacy of the network and the presence of robust interactions among diverse OTUs. Among the eight networks studied, surviving plot B of endophytic networks exhibited the highest edge density of 0.07, while the lowest value of 0.04 was observed in non-surviving plot A of endophytic networks. The highest modularity, a network feature assessing the network’s structure, was identified in surviving plot A (12.04) of rhizospheric networks. Conversely, the lowest modularity was observed in surviving plot B of endophytic networks (Figure 5 and Table 1). In summary, the examination of co-occurrence patterns and network centrality analyses allowed us to recognize the intricate nature of operational taxonomic units (OTUs) within each network category.

### 3.3. Enhancement of Soybean Drought Tolerance Through Seed Treatment of Soybean-Associated Bacteria Isolated from a Drought-Conditioned Field

Bacterial seed treatments have emerged as a promising strategy to enhance plant growth and resilience under abiotic stress conditions, particularly drought [3,36]. In this study, we evaluated the growth-promoting characteristics of various bacterial isolates and their effects on soybean performance, revealing significant potential for improving crop yield and stress tolerance. Table 2 and supporting pictures from Appendix A show that most soybean-associated bacterial isolates selected for this study exhibit strong (+++) mucoidal appearance and siderophore production. Nitrogen fixation is generally high across the isolates, with only DRS2, DRS3, and DES5 showing lower levels (+). Phosphate solubilization is positive for most isolates, except for DES2, DES4, and DES5, which show no activity (-). It highlights the variability in growth-promoting traits among the isolates, suggesting potential for their differential use in promoting plant growth under stress conditions. Table 3 shows the identification of 10 bacterial isolates based on 16S rRNA gene sequencing. Six isolates were classified within the *Pseudomonas* genus, exhibiting high levels of sequence similarity (96.37–99.86%) to known sequences of *Pseudomonas* spp., while one isolate each belonged to the genera *Acinetobacter* (99.73%), *Enterobacter* (85.82%), and *Stenotrophomonas* (96.8%). However, isolate DES4 could not be identified due to the lack of homology in the NCBI BLAST database. All identified bacteria were Gram-negative, consistent with their genera.

Table 4 demonstrates substantial enhancements in soybean growth parameters across the various bacterial seed treatments. Specifically, plants treated with DRS2 (*Acinetobacter pittii*) exhibited remarkable growth metrics, including the longest root length (25.58 cm), shoot length (26.01 cm), shoot fresh weight (1.16 g), and shoot dry weight (0.348 g), thereby highlighting the treatment’s efficacy in promoting plant development (Appendix A). Additionally, DRS3 (*Pseudomonas* sp.)-treated plants demonstrated superior root fresh weight (0.194 g) and root water content (79.87%), indicative of improved root development and moisture retention. DRS4 (*Pseudomonas* sp.) treatment also resulted in significant increases in root length, root fresh weight, and root water content. Moreover, DES3 (*Pseudomonas* sp.)-treated plants showed significantly enhanced root length, shoot length, and both root fresh and dry weights. In contrast, untreated control plants consistently exhibited the lowest growth parameters, reinforcing the conclusion that bacterial seed treatments markedly improve soybean growth and enhance water retention capabilities. The results shown in Table 5 indicate the capability of bacterial seed treatments to enhance soybean tolerance to drought stress. Plants treated with DRS2 (*Acinetobacter pittii*) exhibited the lowest drought stress index (3.0), followed by DES2 (*Enterobacter ludwigii*), DRS3 (*Pseudomonas* sp.), DES1 (*Pseudomonas* sp.), and DRS5 (*Pseudomonas* sp.). The untreated control displayed the highest drought stress index (6.6). This observation suggests that bacterial seed treatments contribute to mitigating the negative effects of drought stress on soybean plants. Although the chlorophyll content of the plants with any seed treatments was not significantly different from one another, the numerically higher chlorophyll content observed in treated plants indicates that seed treatment may enhance chlorophyll synthesis under drought stress conditions, albeit to a limited extent.

## 4. Discussion

Drought stress is a major factor influencing agricultural productivity, particularly for crops like soybeans, which are essential to global food systems. The microbial communities associated with plant roots play a crucial role in mediating plant responses to such stress. Our findings reveal that while rhizospheric microbial diversity remains relatively stable under drought conditions, significant differences in endophytic microbial communities are observed between surviving and non-surviving plants. The predominance of beneficial genera such as *Pseudomonas* and *Pantoea* in surviving plants suggests that these bacteria may enhance drought tolerance, whereas the dominance of *Streptomyces* in non-surviving plants could indicate its prevalence in dead and dry plant environments. The predominance of *Pseudomonas* strains in our microbiome analysis is noteworthy, given their potential contributions to enhancing soybean tolerance to severe drought stress. Extensive studies have highlighted the ability of *Pseudomonas* strains to produce volatile organic compounds (VOCs) that directly assist plants in withstanding drought and high salinity [37]. Furthermore, strains of *Pseudomonas*, like *P. simiae* AU, have been linked to induced systemic tolerance (IST) in plants, helping them withstand various abiotic stresses by promoting proline accumulation and decreasing sodium content in roots to manage osmotic and ionic stress [38]. The formation of biofilms by plant-beneficial *Pseudomonas* spp. has also been identified as a mechanism that improves tolerance to various stresses, including osmotic and oxidative stress while enabling the production of beneficial secondary metabolites [39,40]. Additionally, *Pseudomonas fluorescens* DR397, isolated from drought-prone rhizospheric soil, exhibited high metabolic activity under drought conditions and upregulated the expression of genes related to plant growth promotion, resulting in increased shoot and root growth in legume cultivars under drought conditions [41]. Moreover, the application of *Pseudomonas putida* H-2-3 reprogrammed chlorophyll, stress hormones, and antioxidants in abiotic stress-affected soybean plants, improving their growth under saline and drought conditions [42]. Additionally, under normal conditions, *Pseudomonas putida* SABB7 has been found beneficial in promoting soybean growth and yield [43]. Collectively, these findings underline the crucial role of *Pseudomonas* strains in mitigating drought stress in soybeans through various mechanisms, including VOC production, biofilm formation, and plant physiological modulation.

The other predominant bacterial genus *Pantoea,* in the surviving root samples of Plot A is also reminiscent of previous studies showing the biological roles of *Pantoea* spp. in protecting plants from abiotic stresses. The colonization of wheat plants by *Pantoea agglomerans*, known for its exopolysaccharide (EPS) production, positively influenced rhizosphere soil aggregation by increasing the RAS/RT ratio and enhancing the water stability of adhering soil aggregates [44]. In a separate study, *Pantoea* strain LTYR-11ZT, isolated from the leaves of the drought-tolerant plant *Alhagi sparsifolia*, exhibited multiple plant growth-promoting (PGP) traits, improving wheat performance under drought conditions. This strain enhanced soluble sugar accumulation, reduced proline and malondialdehyde levels, and decreased chlorophyll degradation in leaves [45]. Moreover, EPS derived from *Pantoea alhagi* NX-11 demonstrated significant improvements in drought resistance in rice seedlings, increasing fresh weight and relative water content and enhancing various physiological parameters, including total chlorophyll, proline, and soluble sugar content [46]. Additionally, *Pantoea* sp. YSD J2, which was isolated from the leaves of *Cyperus esculentus* L. var. *sativus*, exhibited notable plant growth-promoting characteristics, including indole acetic acid production, siderophores generation, and the ability to solubilize phosphate and potassium [47]. These combined findings support the multifaceted roles of *Pantoea* species in boosting plant growth, improving drought tolerance, and supporting sustainable agriculture through mechanisms such as EPS production and plant growth-promoting traits.

Our observation that the genus *Streptomyces* was solely prevailing over other major bacterial genera is not surprising, regarding that this bacterial genus is one of the most abundant microorganisms in soil [48,49]. In dead root tissues, all the bacterial organisms that are dependent on the interactions with living plant tissues would be rapidly replaced with the dominant soil saprophytes, such as *Streptomyces* spp. Our results suggest a preference for the saprophytic lifestyle of *Streptomyces* in colonizing dead or dry plants, contributing to the intriguing dynamics of microbial interactions in stressed plant environments. Further investigations into the specific mechanisms underlying *Streptomyces*’s preferences and its potential impact on plant health in stressed conditions would be valuable for a comprehensive understanding of its ecological role.

Regarding the other four major bacterial genera identified in this study, several previous studies reported their relatedness with plant drought stress, supporting the idea that these bacterial organisms can be good biological materials to augment the resilience of crops to drought stress. Arun et al. (2012) reported the isolation of *Micrococcus* spp. from various environmental sources, including soil and plant samples, and highlighted the plant growth-promoting properties of *Micrococcus luteus* [50]. Notably, the strain K39 of *Micrococcus luteus*, isolated from the roots of *Cyperus conglomeratus* in a desert environment, exhibited characteristics relevant to drought stress survival [51]. This finding aligns with our results, where Plot A, experiencing severe drought stress, showed a higher abundance of bacteria from the Micrococcaceae family compared to Plot B. Related to another major genus *Lutemonas*, *L. deserti* sp. nov. is an intriguing species isolated from the desert soil of northern PR China, providing evidence for the presence of *Luteimonas* in the endosphere of drought-stressed plants [52]. This discovery aligns with our observations, suggesting a potential association between *Luteimonas* and plants experiencing drought stress.

*Variovorax paradoxus* strain 5 C-2, characterized by the presence of the enzyme 1aminocyclopropane-1-carboxylate (ACC) deaminase, has been associated with notable benefits for plant growth under drought conditions. Belimov et al. reported that this strain contributes to a reduction in ethylene (ET) production, leading to increased nodulation, elevated seed nitrogen content, enhanced xylem abscisic acid (ABA) concentration, improved water content, and ultimately a higher pea yield [53]. The findings suggest that the enzymatic activity of ACC deaminase in *Variovorax paradoxus* plays a crucial role in modulating hormone signaling both locally in the rhizosphere and systemically within the plant. These observations underscore the potential of *Variovorax paradoxus* as a beneficial rhizobacterium in promoting plant growth and yield, particularly in conditions of soil moisture limitation.

*Bacillus*, particularly *Bacillus thuringiensis* (UFGS2), has demonstrated its ability to mitigate the impacts of drought stress in plants. In soybeans, UFGS2-treated plants exhibited higher stomatal conductance and transpiration compared to the control group following drought stress [54]. This suggests a positive influence of *Bacillus thuringiensis* on plant water regulation under water scarcity conditions. Similarly, the combined application of *Pseudomonas putida* and *Bacillus amyloliquefaciens* alleviated drought stress in chickpeas by exhibiting multiple plant growth-promoting traits, including ACC deaminase activity, mineral solubilization, hormone production, biofilm formation, and siderophore production [55]. Moreover, *Bacillus paralicheniformis* strain FMCH001 demonstrated the potential to enhance water use efficiency, nutrient uptake, root growth, photosynthesis rate, C:N ratio, and overall plant–water relations in soybean, making it a promising candidate for sustaining plant growth in water-limited conditions [56]. Additionally, *Bacillus pumilus* strain SH-9, identified as a drought-tolerant variant, positively influenced soybean growth under drought stress by modulating the expression of phytohormone genes and antioxidant profiles [57]. Furthermore, *Bacillus subtilis* emerged as a beneficial bacterium, promoting the growth of common beans and maize while increasing water use efficiency. This bacterium enhanced leaf water content, regulated stomatal activity, and decreased antioxidant activities without compromising photosynthetic rates [58]. In summary, *Bacillus* species, including *Bacillus thuringiensis*, *Bacillus amyloliquefaciens*, *Bacillus paralicheniformis*, *Bacillus pumilus*, and *Bacillus subtilis*, exhibit promising capabilities in alleviating drought stress and enhancing plant growth under challenging environmental conditions. These findings emphasize the potential of *Bacillus*-based strategies for sustainable agriculture in water-limited regions.

The findings from the greenhouse experiment indicate that bacterial seed treatments substantially enhance soybean growth and drought tolerance, highlighting the potential of microbial applications in sustainable agriculture. Isolates such as *Acinetobacter pittii* (DRS2) and *Pseudomonas* species (e.g., DRS3 and DES3) consistently demonstrated superior root and shoot growth, along with increased water retention under drought stress. These isolates help crops tolerate drought stress through various mechanisms, including the production of exopolysaccharides, volatile compounds, accumulation of osmolytes, up- or downregulation of stress-responsive genes, and changes in root morphology [59,60]. This improvement likely stems from the isolates’ growth-promoting traits, including nitrogen fixation, siderophore production, and phosphate solubilization, as evidenced by significant increases in plant height, biomass, and water content in treated groups compared to the control. The superior performance of these isolates may be attributed to their ability to enhance water uptake, improve nutrient acquisition, boost stress tolerance, and modulate hormonal signaling pathways [42,60].

Additionally, the reduced drought stress index (DSI) and enhanced chlorophyll content observed in treated plants support the role of these bacterial isolates in maintaining physiological functions under limited water availability. This finding aligns with studies highlighting *Enterobacter* and other desiccation-tolerant plant growth-promoting rhizobacteria (PGPR) as beneficial in such conditions [61]. Despite the limited differences in chlorophyll content, the numerically higher SPAD readings suggest potential chlorophyll stability under stress, possibly associated with bacterial influence on nutrient availability and root health [42]. These results align with previous research emphasizing the role of plant-associated bacteria in drought mitigation, positioning these isolates as viable biofertilizers for improving drought resilience in soybeans [54,62]. Future work should focus on the field application of these isolates across diverse environmental conditions to validate their effectiveness and adaptability. Moreover, future research should explore the underlying mechanisms of action, optimize application strategies, and evaluate the long-term impacts of these treatments on soybean yield and quality.

## 5. Conclusions

This study contributes to a deeper understanding of the intricate interactions between bacteria and plants in the context of drought stress. Specifically, our metagenomic analyses in the fields of Louisiana under drought conditions have provided insights into microbial diversity and primary bacterial components of the rhizosphere and endosphere under such arid circumstances. The findings from our co-occurrence network analyses have corroborated and strengthened our understanding of microbial dynamics in response to drought stress. Additionally, results from greenhouse experiments supported the growth-promoting characteristics and drought stress mitigation potential of rhizospheric and endospheric bacterial isolates in soybean plants. Our study establishes a foundation for further research and highlights the importance of defining microbial diversity in the context of drought stress, particularly in regions like Louisiana facing challenges related to drought conditions.

## Figures and Tables

**Figure 1 microorganisms-12-02630-f001:**
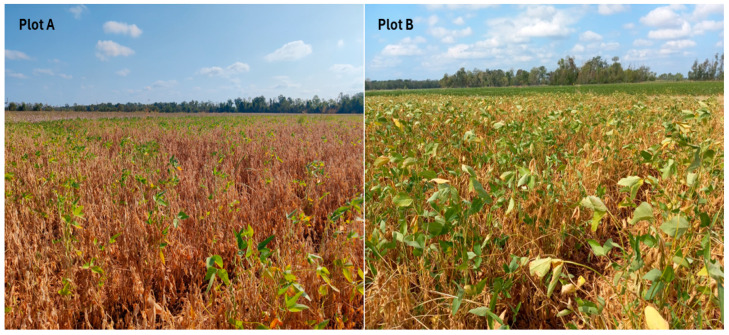
The drought-damaged soybean field where the plant samples were collected (Alexandria, Louisiana). Plot A and Plot B neighbored each other.

**Figure 2 microorganisms-12-02630-f002:**
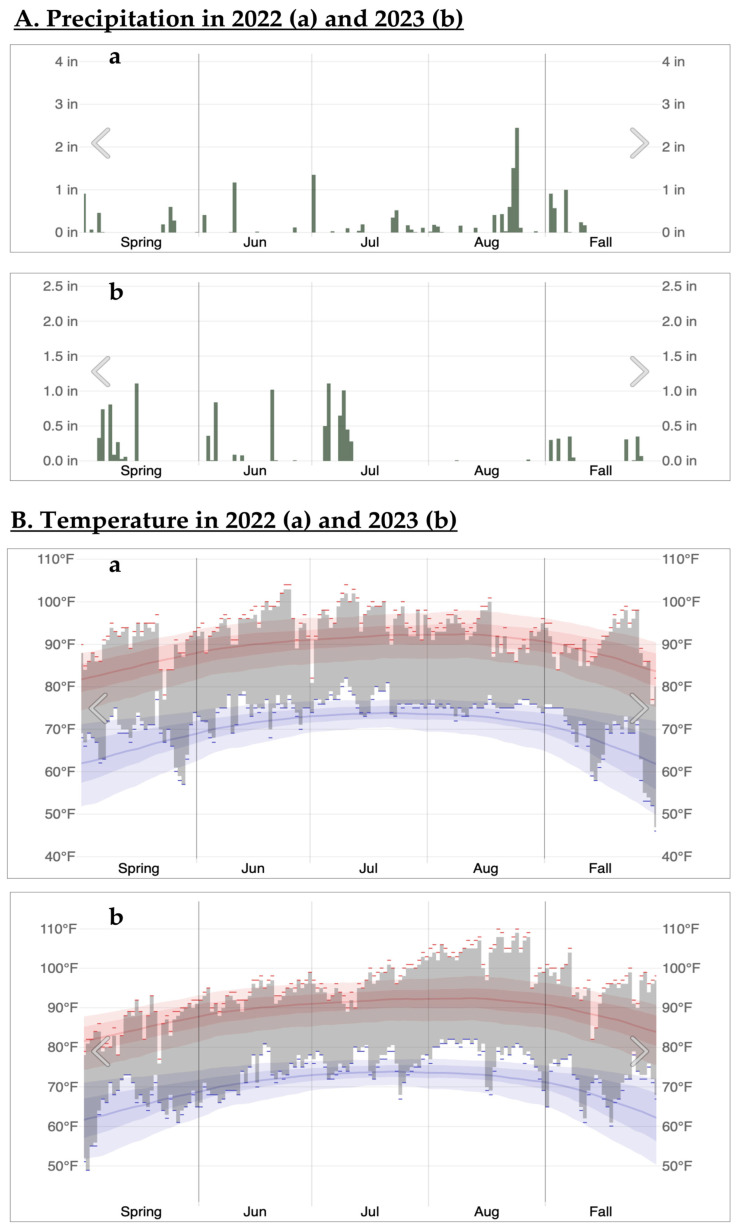
The records of rainfalls (**A**) and temperatures (**B**) in Alexandria, Louisiana, during the 2022 and 2023 growing seasons. These graphs were obtained from Weather Spark (https://weatherspark.com) (data source: The Alexandria International Airport, accessed on 4 January 2024).

**Figure 3 microorganisms-12-02630-f003:**
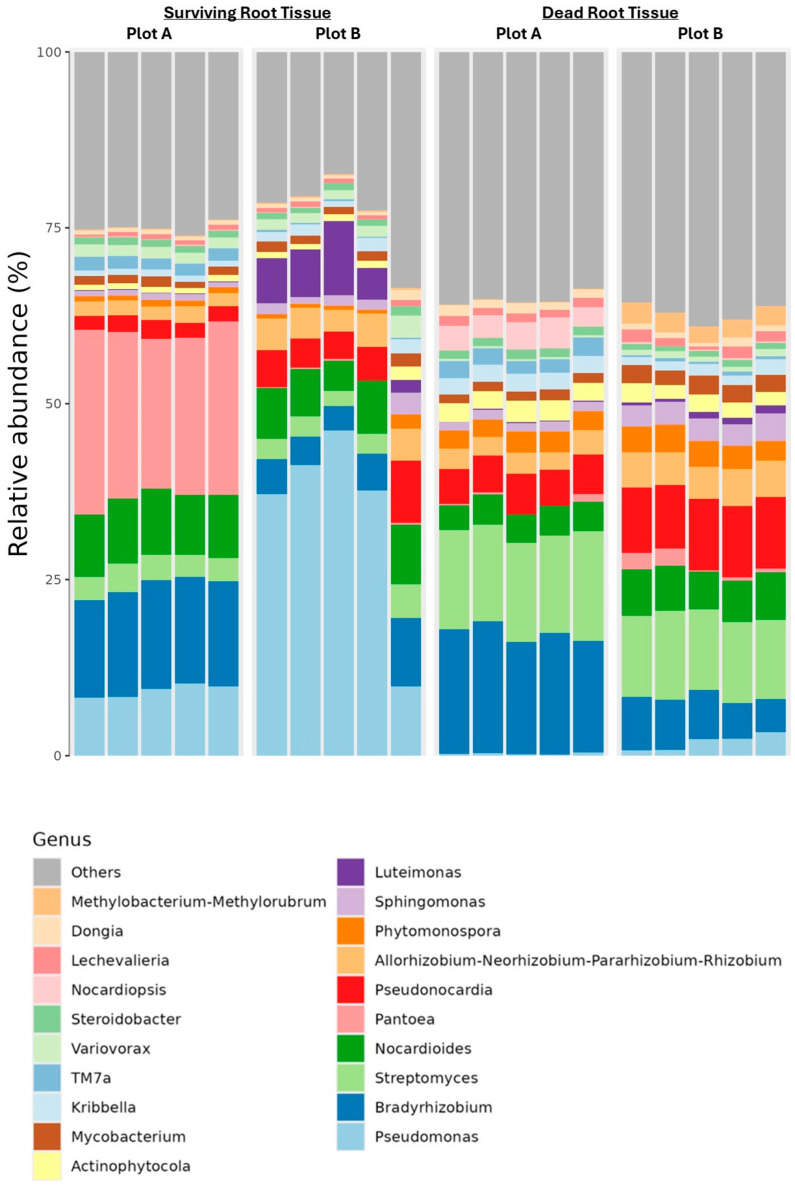
Taxa bar plot showing seven abundant bacterial genera either in surviving or dry/dead soybean root tissues (root endosphere).

**Figure 4 microorganisms-12-02630-f004:**
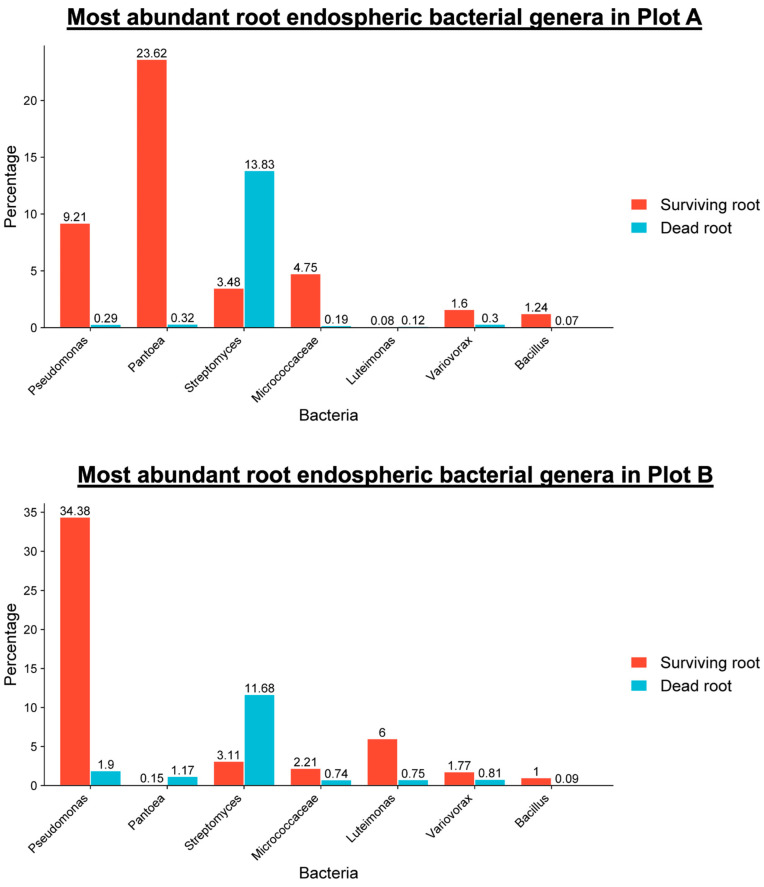
Relative abundance of bacterial genera dominant in the root endosphere of drought-damaged soybean plants.

**Figure 5 microorganisms-12-02630-f005:**
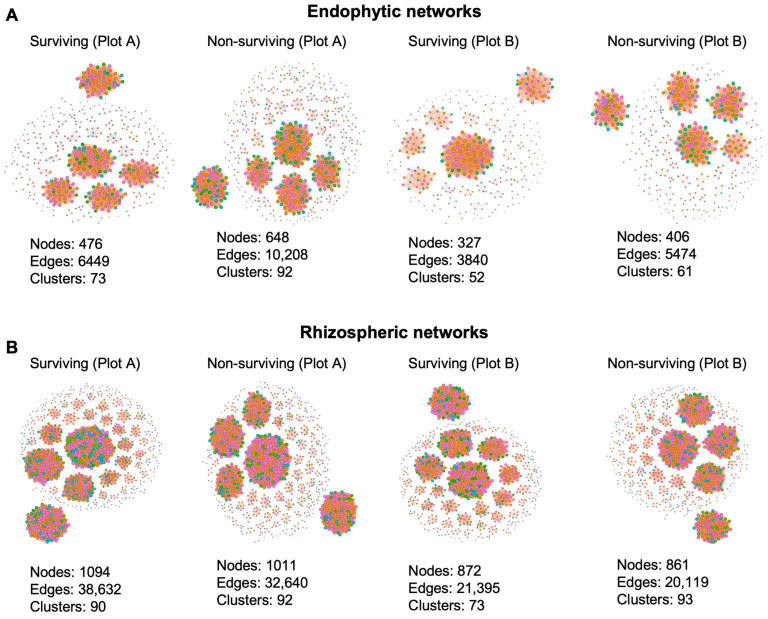
The co-occurrence networks analyses. Four groups each for endophytic (**A**) and rhizospheric networks (**B**) are illustrated. Individual types of networks within each category are indicated. Nodes, edges, and clusters for each network are specified. The color of nodes signifies OTUs from the same module in each network, while line color indicates positive (orange) and negative (blue) correlation coefficients. Network construction employed Spearman’s correlation coefficient, with R > 0.9 and *p* < 0.05 as criteria.

**Table 1 microorganisms-12-02630-t001:** Network features of diverse co-occurrence networks.

		Endophytic Network			Rhizospheric Network			
Centrality Features	Surviving (Plot A)	Non-Surviving (Plot A)	Surviving (Plot B)	Non-Surviving (Plot B)	Surviving (Plot A)	Non-Surviving (Plot A	Surviving (Plot B)	Non-Surviving (Plot B)
Nodes	476	648	327	406	1094	1011	872	861
Edges	6449	10208	3840	5474	38632	32640	21395	20119
Positive Edges	6320	10050	3765	5426	38590	32543	21352	19992
Negative Edges	129	158	75	48	42	97	43	127
Number of Clusters	73	92	52	61	90	92	73	93
Connectance (Edge Density)	0.05704556	0.048695785	0.07204368	0.06658152	0.06461595	0.063930429	0.0563388	0.054341896
Average degree (Average K)	27.0966387	31.50617284	23.4862385	26.9655172	70.6252285	64.56973294	49.071101	46.7340302
Average Path Length	1	1	1	1	1	1	1	1
Diameter	1	1	1	1	1	1	1	1
Mean Clustering Coefficient (Average CC)	1	1	1	1	1	1	1	1
Centralization Degree	0.06927023	0.064132654	0.1334778	0.06181354	0.08817454	0.095475512	0.0791377	0.0712395
RM (Relative Modularity)	5.2693558	6.040679146	3.17505499	4.79215729	12.0494888	10.78020977	8.7625972	8.561277694

**Table 2 microorganisms-12-02630-t002:** Growth-promoting characteristics of selected bacterial isolates.

Bacterial Isolates	Mucoidal Appearance	Siderophore Production	Nitrogen Fixation	Phosphate Solubilization
DRS1	+++	+++	+++	+++
DRS2	++	+++	+	+++
DRS3	+++	+++	+	+++
DRS4	+++	+++	+++	+++
DRS5	+++	+++	+++	+
DES1	+++	+++	+++	+
DES2	+++	+++	+++	−
DES3	+++	+++	+++	++
DES4	+++	+++	+++	−
DES5	++	+	+	−

+++: Positive. ++: Positive but low. +: Positive but very low. −: Negative.

**Table 3 microorganisms-12-02630-t003:** Identification of bacteria isolated from the soybean rhizosphere and endosphere based on the 16S rRNA gene sequence.

Isolate	Gram +/−	Probable Identity with Highest Homology Match	% Similarity
DRS1	−	*Pseudomonas lini*	99.18
DRS2	−	*Acinetobacter pittii*	99.73
DRS3	−	*Pseudomonas* sp.	98.63
DRS4	−	*Pseudomonas* sp.	96.37
DRS5	−	*Pseudomonas* sp.	99.86
DES1	−	*Pseudomonas* sp.	97.75
DES2	−	*Enterobacter ludwigii*	85.82
DES3	−	*Pseudomonas* sp.	99.33
DES4	*	*	No similarity found
DES5	−	*Stenotrophomonas* sp.	96.8

*: Unidentified bacterial strain.

**Table 4 microorganisms-12-02630-t004:** Effects of bacterial seed treatments on seedling growth parameters of soybean under drought stress conditions.

Bacterial Seed Treatments	Root Length (cm)	Shoot Length (cm)	Root Fresh Weight (g)	Shoot Fresh Weight (g)	Root dry Weight (g)	Shoot Dry Weight (g)	Root Water Content (%)	Shoot Water Content (%)
DRS1	20.63 ^cd^	24.30 ^ab^	0.150 ^bcde^	0.77 ^bcd^	0.051 ^cde^	0.253 ^bcd^	66.37 ^bcd^	66.69 ^bc^
DRS2	25.58 ^a^	26.01 ^a^	0.188 ^ab^	1.16 ^a^	0.072 ^ab^	0.348 ^a^	60.82 ^cde^	69.48 ^abc^
DRS3	23.49 ^ab^	23.67 ^abc^	0.194 ^a^	0.71 ^cd^	0.038 ^e^	0.218 ^cde^	79.87 ^a^	68.04 ^abc^
DRS4	22.80 ^abc^	24.16 ^ab^	0.204 ^a^	0.97 ^ab^	0.056 ^bcd^	0.298 ^ab^	72.23 ^ab^	68.81 ^abc^
DRS5	19.86 ^d^	23.74 ^abc^	0.172 ^abcd^	0.69 ^d^	0.045 ^cde^	0.212 ^cde^	74.10 ^ab^	67.75 ^abc^
DES1	19.06 ^d^	24.70 ^ab^	0.139 ^cde^	0.58 ^d^	0.052 ^cde^	0.204 ^cde^	63.02 c^de^	64.14 ^bc^
DES2	20.46 ^cd^	20.95 ^d^	0.134 ^de^	0.61 ^d^	0.040 ^de^	0.226 ^cde^	68.79 ^bc^	61.38 ^c^
DES3	24.74 ^a^	24.19 ^ab^	0.181 ^abc^	0.94 ^abc^	0.079 ^a^	0.262 ^bc^	55.52 ^e^	71.65 ^ab^
DES4	20.06 ^cd^	23.47 ^bc^	0.148 ^bcde^	0.58 ^d^	0.037 ^e^	0.180 ^e^	74.20 ^ab^	67.96 ^abc^
DES5	21.15 ^bcd^	24.77 ^ab^	0.108 ^e^	0.77 ^bcd^	0.039 ^de^	0.221 ^cde^	62.53 ^cde^	70.16 ^ab^
CMC	18.81 ^d^	23.80 ^abc^	0.146 ^bcde^	1.01 ^ab^	0.060 ^bc^	0.246 ^bcde^	58.69 ^de^	75.64 ^a^
Utr	18.76 ^d^	21.69 ^cd^	0.118 ^e^	0.65 ^d^	0.045 ^cde^	0.188 ^de^	60.01 ^cde^	70.51 ^ab^
Treatment effect (ANOVA)	*p* < 0.0001							

Means followed by different letters are significantly different from each other at a significance level of *p* < 0.05.

**Table 5 microorganisms-12-02630-t005:** Effects of bacterial seed treatments on drought stress tolerance and chlorophyll content of soybeans under drought stress conditions.

Bacterial Seed Treatments	Drought Stress Index (DSI)	SPAD Reading
DRS1	6.0 ^a^	37.8 ^a^
DRS2	3.0 ^b^	39.6 ^a^
DRS3	4.4 ^ab^	38.4 ^a^
DRS4	5.6 ^a^	38.4 ^a^
DRS5	5.4 ^ab^	39.6 ^a^
DES1	4.8 ^ab^	35.2 ^a^
DES2	4.2 ^ab^	35.7 ^a^
DES3	5.6 ^a^	36.4 ^a^
DES4	5.8 ^a^	35.7 ^a^
DES5	6.2 ^a^	36.9 ^a^
CMC	6.2 ^a^	38.7 ^a^
Utr	6.6 ^a^	36.0 ^a^
Treatment effect (ANOVA)	*p* < 0.0001	*p* = 0.045

Means followed by different letters are significantly different from each other at a significance level of *p* < 0.05.

## Data Availability

The data supporting the results of this study can be obtained from the corresponding authors upon reasonable request. The DNA sequence data analyzed in this study were deposited in the National Center for Biotechnology Information (NCBI) database (BioProject ID: PRJNA1193106).

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
