# Peer review of "Microbiome Structures and Beneficial Bacteria in Soybean Roots Under Field Conditions of Prolonged High Temperatures and Drought Stress"

_microorganisms, 2024, doi:10.3390/microorganisms12122630_

Round 1
Reviewer 1 Report
Comments and Suggestions for Authors
The reviewed MS dedicated to the study of the microbiomes in soybean roots under drought conditions. This study is important for microbiology and agricultural biotechnology. The paper is very clear; it will be of interest to a wide audience. The mostly recent publications were cited. The modern methods of metagenomics analysis, and appropriate statistical approaches were used. It is necessary to note, that the figures and tables are uncomplicated and illustrate the basic ideas of the research. The experimental data interpreted consistently throughout the manuscript. The conclusions of the MS supported by the results. It is necessary to note, that all parts of the MS are well edited.
I can recommend the MS for the publication after some minor corrections.
Suggestions to the authors
Keywords: Please, try not to use the terms from the title as keywords.
Lines 82-90: In this part of the introduction the methods were described. I think, that it is better to transfer these sentences at the beginning of Section 2.1.
Figure 2: Please improve the image quality. Maybe you could enlarge it.
Line 233: You mentioned Figures S1, but I wasn’t able to find it.
Author Response
The reviewed MS dedicated to the study of the microbiomes in soybean roots under drought conditions. This study is important for microbiology and agricultural biotechnology. The paper is very clear; it will be of interest to a wide audience. The mostly recent publications were cited. The modern methods of metagenomics analysis, and appropriate statistical approaches were used. It is necessary to note, that the figures and tables are uncomplicated and illustrate the basic ideas of the research. The experimental data interpreted consistently throughout the manuscript. The conclusions of the MS supported by the results. It is necessary to note, that all parts of the MS are well edited.
I can recommend the MS for the publication after some minor corrections.
Response: We greatly appreciate your positive and supportive comments on our manuscript.
Suggestions to the authors
Keywords: Please, try not to use the terms from the title as keywords.
Response: Key words have been changed to ‘soybean rhizosphere’, ‘root endosphere’, and ‘microbial community’.
Lines 82-90: In this part of the introduction the methods were described. I think, that it is better to transfer these sentences at the beginning of Section 2.1.
Response: We moved that part to Section 2.1 as suggested (Line 100 – 113 in the revision).
Figure 2: Please improve the image quality. Maybe you could enlarge it.
Response: We enlarged Figure 2 as suggested.
Line 233: You mentioned Figures S1, but I wasn’t able to find it.
Response: Figure S1, as well as Figures S2-S4, are included in the file, ‘Supplementary Information’. To avoid confusion, we added ‘in Supplementary Information’ when Figures S1-S4 are cited in the text of the revision.

Reviewer 2 Report
Comments and Suggestions for Authors
The manuscript entitled „Microbiome Structures and Beneficial Bacteria in Soybean Roots under Field Conditions of Prolonged High Temperatures and Drought Stress” presents studies concentrated in the area of plant microbiology. Applying the next generation sequencing make it novel, significant for the development of science and interesting to the broad audience.
The manuscript is well structured, data are well presented and discussion is well written, however I have some suggestions for improvement the quality of paper.
- Abstract section. Please use passive voice to emphasize the research, especially when the agent of activity is obvious to the readers e.g. The endophytic and rhizospheric microbial diversity analyses were conducted” instead „We conducted endophytic and rhizospheric microbial diversity analyses”
- Introduction section – please mention more about the importance of the microbiome for the plant response to drought stress. Moreover, I suggest transferring Figure 1 into the materials and methods section and remove the lines 97-106 (it is rather a summary not an introduction). L90-93 – transfer it to the testable hypothesis and refer to this hypothesis in the discussion section. Avoid words that are very general like „investigate” or „analyze”.
- Materials and methods section – L163 – please provide more details about DNA isolation from endosphere samples.
- To make the results more reliable please provide the raw sequences data (e.g. SRA NCBI database)
Author Response
The manuscript entitled „Microbiome Structures and Beneficial Bacteria in Soybean Roots under Field Conditions of Prolonged High Temperatures and Drought Stress” presents studies concentrated in the area of plant microbiology. Applying the next generation sequencing make it novel, significant for the development of science and interesting to the broad audience.
The manuscript is well structured, data are well presented and discussion is well written, however I have some suggestions for improvement the quality of paper.
Response: We greatly appreciate your positive and supportive comments on this manuscript.
Abstract section. Please use passive voice to emphasize the research, especially when the agent of activity is obvious to the readers e.g. The endophytic and rhizospheric microbial diversity analyses were conducted” instead „We conducted endophytic and rhizospheric microbial diversity analyses”
Response: We changed the sentences to passive form in the abstract section, except for the sentence describing the activity for isolation and identification of beneficial bacteria (Lines 25, 27, 29, and 30 in the revision).
Introduction section – please mention more about the importance of the microbiome for the plant response to drought stress.
Response: More information about related previous studies is added (Lines 61-78 in the revision).
Moreover, I suggest transferring Figure 1 into the materials and methods section and remove the lines 97-106 (it is rather a summary not an introduction).
Response: Figure 1 was moved to the materials and methods section. The sentences on line 97-106 were removed as suggested.
L90-93 – transfer it to the testable hypothesis and refer to this hypothesis in the discussion section. Avoid words that are very general like „investigate” or „analyze”.
Response: The corresponding sentence was changed to “Additionally, we isolated and identified soybean-associated bacteria from drought-stressed soybean plants and tested their beneficial effects on soybean growth under dought stress with the hypothesis that those bacteria would enhance the drought tolerance of soybean plants.” (Lines 92-95 in the revision). Results of experiments based on this hypothesis are discussed in the discussion section (Lines 602-615 in the revision).
Materials and methods section – L163 – please provide more details about DNA isolation from endosphere samples.
Response: Detailed procedure for endosphere sampling is described in Lines 164-172 in the revision.
To make the results more reliable please provide the raw sequences data (e.g. SRA NCBI database)
Response: The DNA sequence data used in this study were deposited in the NCBI database, and the ‘Data Availability Statement’ was modified accordingly (Lines 655-658 in the revision).
